

**Land Surface Model Representation of the Mutual Information**
**Context between Multi-Layer Soil Moisture and**
**Evapotranspiration**
Jianxiu Qiu[1,2], Wade T. Crow[3], Jianzhi Dong[3], Grey S. Nearing[4]
[1]Guangdong Provincial Key Laboratory of Urbanization and Geo-simulation, School of Geography and Planning, Sun
Yat-sen University, Guangzhou, 510275, China
[2]Southern Laboratory of Ocean Science and Engineering (Guangdong, Zhuhai), Zhuhai, 519000, China
[3]USDA ARS Hydrology and Remote Sensing Laboratory, Beltsville, MD 20705, USA
[4]Department of Geological Sciences, University of Alabama, AL 35487, USA
*Correspondence to*: Jianxiu Qiu (qiujianxiu@mail.sysu.edu.cn)



**Abstract.** Soil moisture ($\theta$) impacts the climate system by regulating incoming energy into outgoing evapotranspiration (ET) and sensible heat flux components. Therefore, investigating the coupling strength between $\theta$ and ET is important for the study of land surface/atmosphere interactions. Here, we use in-situ AmeriFlux observations to evaluate $\theta$/ET coupling strength estimates acquired from multiple land surface models (LSMs). For maximum robustness, coupling strength is represented using the sampled normalized mutual information (NMI) between $\theta$ estimates acquired at various vertical depths and surface flux represented by fraction of potential evapotranspiration (fPET, the ratio of ET to potential ET). Results indicate that LSMs are generally in agreement with AmeriFlux measurements in that surface soil moisture ($\theta_S$) contains slightly more NMI with fPET than vertically integrated soil moisture ($\theta_V$). Overall, LSMs adequately capture variations in NMI between fPET and $\theta$ estimates acquired at various vertical depths. However, one model – the Global Land Evaporation Amsterdam Model (GLEAM) – significantly overestimates the NMI between $\theta$ and ET and the relative contribution of $\theta_S$ to total ET. This bias appears attributable to differences in GLEAM's ET estimation scheme relative to the other two LSMs considered here (i.e., the Noah with Multi–parameterization option and the Catchment Land Surface Model). These results provide insight into improved LSM model structure and parameter optimization for land surface-atmosphere coupling analyses.

**Keywords.** Land surface/atmosphere interaction, soil moisture, surface evapotranspiration

## 1 Introduction

Soil moisture ($\theta$) modulates water and energy feedbacks between the land surface and the lower atmosphere by partitioning incoming energy into evapotranspiration (ET) and sensible heat (H) surface flux components (Seneviratne et al., 2010, 2013). In water-limited regimes, $\theta$ exhibits a dominant control on ET and, therefore, commonly exerts significant terrestrial control on the earth's water, energy and biochemical cycles. Accurately representing $\theta$/ET coupling in land surface models (LSMs) is therefore expected to improve our ability to project the future frequency of extreme climates (Seneviratne et al., 2013).

A key question is how the constraint of $\theta$ on ET and H varies as $\theta$ is vertically integrated over deeper vertical soil depths. Given the tendency for the time scales of $\theta$ dynamics to vary strongly with depth, the degree to which the ET is coupled with vertical variations in $\theta$ determines the temporal scale at which $\theta$ variations are propagated into the atmosphere. Therefore, in order to represent $\theta$/ET coupling, and thus land/atmosphere interactions in general, LSMs must accurately capture the relationship between vertically varying $\theta$ values and ET. Unfortunately, their abilities to do so remains an open question.

Recently, land surface/atmosphere coupling strength has been investigated by sampling mutual information proxies (e.g., correlation coefficient or other coupling indices) between time series of $\theta$ and ET (or air temperature proxies for ET). Results suggest that, even when confined to very limited vertical support (e.g., within the top 5 cm of the soil column), surface $\theta$ estimates retain significant information for examining $\theta$ controls on local climate (Ford and Quiring, 2014b; Qiu et al., 2014; Dong and Crow, 2018; Dong and Crow, 2019). These findings are in contrast with the common



perceptions that ET is dominated by $\theta$ values at deeper soil layers (Hirschi et al., 2014). Hence, it is necessary to
examine whether LSMs can realistically reflect observed variations of $\theta$/ET coupling strength in the vertical soil
profiles.
Previous studies examining the $\theta$/ET relationship have generally been based on Pearson product-moment correlation
(Basara and Crawford, 2002; Ford et al., 2014a), which captures only the strength of a linear relationship between two
variables. However, the coupling between $\theta$ and ET is generally nonlinear. Therefore, non-parametric mutual
information measures are generally more appropriate. Nearing et al. (2018) used information theory metrics (transfer
entropy, in particular) to measure the strengths of directed couplings between different surface variables, including
soil moisture, and surface energy fluxes at short timescales in several LSMs. They found that the LSMs were generally
biased as compared with strengths of couplings in observation data, and that these biases differed across different
study sites. However, they did not look at the effect of vertical moisture profiles or of subsurface soil moisture on
partitioning surface energy fluxes.
Here we apply the information theory-based methodology of Qiu et al. (2016) to examine the relationship between
the vertical support of $\theta$ estimates and their mutual information (MI) with respect to ET. Our approach is based on
analyzing the MI content between ET and $\theta$ time series - acquired from both LSMs and AmeriFlux in-situ observations.
MI values are then normalized by corresponding ET entropy to remove the effect of inter-site variation and generate
estimates of Normalized Mutual Information (NMI) between $\theta$ and ET. Examined $\theta$ time series have two different
vertical supports: surface soil moisture ($\theta_S$) and vertically integrated soil moisture ($\theta_V$). AmeriFlux NMI results are
compared with analogous NMI results obtained from LSM-based $\theta$ and ET time series.
Further details on our methodology are presented in Sect. 2. Results are presented in Sect. 3 and discussed/summarized
in Sect. 4.

**2 Data and Methods**

The AmeriFlux network provides temporally continuous measurements of $\theta$, surface energy fluxes and related
environmental variables for sites located in a variety of North American ecosystem types, e.g., forests, grasslands,
croplands, shrublands and savannas (Boden, et al., 2013). To minimize sampling errors, AmeriFlux sites lacking a
complete 3-year summer months (June, July and August) daily time series between the years of 2003 and 2015 (i.e.,
$3 \times 92 = 276$ daily observations in total) of $\theta_S$, $\theta_V$ and latent heat flux (LE) were excluded here - resulting in the 34
eligible AmeriFlux sites listed in Table 1. These sites cover a variety of climate zones within the contiguous United
States (CONUS). Table 1 gives background information on these 34 sites including local land cover information.
Hydro-climatic conditions in each site were characterized using the aridity index (AI) – calculated using CRU (Climate
Research Unit, v4.02) monthly precipitation and potential evaporation (PET) datasets.
As described above, $\theta$/ET coupling assessments made using AmeriFlux observations were compared with comparable
assessments based on output from state-of-the-art LSMs including Noah with Multi–parameterization option



(NOAHMP), Catchment Land Surface Model (CLSM), and Global Land Evaporation Amsterdam Model (GLEAM).
See below for more model details. To avoid any spurious correlations between $\theta$ and ET due to seasonality, all NMI
analyses were performed on $\theta$ and ET time series anomalies acquired during the period 2003–2015. The $\theta$ and ET
anomalies were calculated by removing the seasonal cycle – defined as 31-day window averages centered on each
day-of-year sampled across all years of the 2003–2015 historical data record – from the raw $\theta$ and ET time series data.
The analysis was limited to the CONUS during summer months (June, July and August) when $\theta$/ET coupling was
expected to be maximized.
Table 1 Attributes of selected AmeriFlux sites

| AmeriFlux sites | Land cover | Elevation [m] | Top-layer depth [cm] | Bottom-layer depth [cm] |
|---|---|---|---|---|
| ARM SGP Main | Cropland | 314 | 10[a] | 20[b] |
| ARM USDA UNL OSU Woodward Switchgrass 1 | Grassland | 611 | 10 | 30 |
| Audubon Research Ranch | Grassland | 1469 | 10 | 20 |
| Bondville | Cropland | 219 | 10[c] | 20 |
| Brookings | Grassland | 510 | 10 | 20 |
| Chimney Park | Evergreen needleleaf forest | 2750 | 0-15 | 15-45 |
| Duke Forest Hardwoods | Deciduous broadleaf forest | 168 | 10 | 25 |
| Duke Forest Open Field | Grassland | 168 | 10 | 25 |
| Fermi Agricultural | Cropland | 225 | 2.5 | 10 |
| Fermi Prairie | Grassland | 226 | 2.5 | 10 |
| Flagstaff Managed Forest | Evergreen needleleaf forest | 2160 | 2 | 10 |
| Flagstaff Unmanaged Forest | Woody savannas | 2180 | 2 | 10 |
| Flagstaff Wildfire | Grassland | 2270 | 2 | 10 |
| Fort Peck | Grassland | 634 | 5[d] | 20 |
| Freeman Ranch Woodland | Woody savannas | 232 | 10 | 20 |
| Glacier Lakes Ecosystem Experiments Site | Evergreen needleleaf forest | 3190 | 5 | 10 |
| Howland Forest Main | Mixed forest | 60 | NA | NA |
| Lucky Hills Shrubland | Open shrubland | 1372 | 5 | 15 |
| Marys River Fir Site | Evergreen needleleaf forest | 263 | 10 | 20 |
| Metolius Intermediate Pine | Evergreen needleleaf forest | 1253 | 0-30 | NA |
| Missouri Ozark | Deciduous broadleaf forest | 219 | 10 | 100 |
| Nebraska SandHills Dry Valley | Grassland | 1081 | 10 | 25 |
| Quebec Boreal Cutover Site | Evergreen needleleaf forest | 400 | 5 | 20 |
| Quebec Mature Boreal Forest Site | Evergreen needleleaf forest | 400 | 5 | 10 |
| Santa Rita Creosote | Open shrubland | 991 | 2.5 | 12.5 |
| Santa Rita Mesquite | Woody savannas | 1116 | 2.5-5 | 5-10 |
| Sherman Island | Grassland | -5 | 10 | 20 |
| Sylvania Wilderness | Mixed forest | 540 | 5 | 10 |
| Tonzi Ranch | Woody savannas | 169 | 0 | 20 |





| | | | | |
|---|---|---|---|---|
| University of Michigan Biological Station | Deciduous broadleaf forest | 234 | 0-30 | NA |
| Vaira Ranch | Grassland | 129 | 0 | 10 |
| Walker Branch | Deciduous broadleaf forest | 343 | 5 | 10 |
| Willow Creek | Deciduous broadleaf forest | 515 | 5 | 10 |
| Wind River Field Station | Evergreen needleleaf forest | 371 | 30[e] | 50[f] |


[a] Was 5 cm prior to 4/13/2005
[b] Was 25 cm prior to 4/13/2005
[c] Was 5 cm prior to 1/1/2006
[d] Was 10 cm (2003-2008)
[e] Was 0-30 cm prior to 2007
[f] Was NaN prior to 2007
**2.1 Ground-based AmeriFlux measurements**
The Level 2 (L2) AmeriFlux LE and sensible heat (H) flux observations are based on high-frequency (typically > 10
Hz) eddy covariance measurements processed into half-hourly averages by individual AmeriFlux investigators. LE
and $\theta$ observations at a half-hour time step and without gap-filling procedures were collected from the AmeriFlux Site
and Data Exploration System (see http://ameriflux.ornl.gov/). The LE and $\theta$ observations were further aggregated into
daily (0 to 24 UTC) values, and daily LE was converted into daily ET using the latent heat of vaporization. Daily ET
values based on less than 30% half-hourly coverage (i.e., < 15 half-hourly observations per day) were considered not
representative at a daily time scale and therefore excluded.
Soil moisture measurements are generally available at two discrete depths that vary between the AmeriFlux sites
(Table 1). Here, the top (i.e., closest to the surface) soil moisture observation was always used to represent surface
soil moisture ($\theta_S$). Since the depth of this top layer measurement varies between 0 and 15 cm (see Table 1), we consider
the surface-layer measurement $\theta_S$ to be roughly representative of 0–10 cm (vertically integrated) $\theta$.
Given variations in the depth of the lower AmeriFlux $\theta$ observations (see Table 1), we applied a variety of approaches
for estimating vertically integrated soil moisture ($\theta_V$). Our first approach, hereinafter referred to as Case I, was based
on the application of an exponential filter (Wagner et al., 1999; Albergel et al., 2008) to extrapolate $\theta_S$ to a consistent
40 cm bottom layer depth. Therefore, only $\theta_S$ was used to derive $\theta_V$ and the bottom-layer AmeriFlux $\theta$ measurement
was neglected in this case. The application of the exponential filter requires a single time-scale parameter $T$. Since $\theta$
measurements from United States Department of Agriculture's Soil Climate Analysis Network (SCAN) are taken at
fixed soil depth, we utilized this dataset to determine the most appropriate parameter $T$ at AmeriFlux sites. Following
Qiu et al. (2014), first, we estimated the optimal parameter $T$ ($T$opt) for the extrapolation of $\theta$ measurements from 10
cm to 40 cm depth and established a global relationship between $T$opt and site-based NDVI (MOD13Q1 v006, 250m,
16-day) ($T$opt $= 2.098 \times \exp(-1.895 \times (\mathrm{NDVI} + 0.6271)) + 2.766$). Then, this global relationship (Goodness of Fit $R^2$:
0.85) was applied to AmeriFlux sites to extrapolate 0–10 cm $\theta_S$ times series into 0–40 cm $\theta_V$.


Previous research has suggested that such a filtering approach does not significantly squander ET information present
in actual measurements of $\theta_V$ (Qiu et al., 2014; Qiu et al., 2016). Nevertheless, since the quality of $\theta_V$ estimates is
important in our analysis, we also calculated two addition cases where 0–40 cm $\theta_V$ was estimated using: 1) the bottom-
layer soil moisture measurement acquired at each AmeriFlux site (hereinafter, Case II) and 2) linear interpolation of
$\theta_S$ and the bottom-layer AmeriFlux soil moisture measurement (hereinafter, Case III). The sensitivity of key results to
these various cases is discussed below.
**2.2 LSM-based simulations**
LSM output was acquired from NOAHMP (Niu et al., 2011) and CLSM (Koster et al., 2000) simulations embedded
within the NASA Land Information System (LIS, Kumar et al., 2006) and a satellite-observation-based model
GLEAM (Miralles et al., 2011). Both NOAHMP and CLSM were set-up to simulate 0.125 °$\theta$ profiles at a 15-minute
time step using North America Land Data Assimilation System, Phase 2 (NLDAS-2) forcing data. A 10-year model
spin-up period (1992 to 2002) was applied for NOAHMP and CLSM.
NOAHMP numerically solves the one-dimensional Richards equation within four soil layers of thicknesses of 10, 30,
60, and 100 cm. Major parameterization options relevant to $\theta$ simulation include: 1) options for canopy stomatal
resistance; 2) options for $\theta$ factor for stomatal resistance (the $\beta$ factor). Here we employed the Ball-Berry-type stomatal
resistance scheme and Noah-type soil moisture factor controlling the $\beta$ factor. The specific expressions are as follows:
$$\beta = \sum_{i=1}^{N_{\mathrm{root}}} \frac{\Delta Z_i}{Z_{root}} \min\left(1.0, \frac{\theta_i - \theta_{\mathrm{wilt}}}{\theta_{\mathrm{ref}} - \theta_{\mathrm{wilt}}}\right) \tag{1}$$

where $\theta_{\mathrm{wilt}}$ and $\theta_{\mathrm{ref}}$ are respectively soil moisture at witling point (m$^3$ m$^{-3}$) and reference soil moisture (m$^3$ m$^{-3}$), which
is close to field capacity. $\theta_i$ and $\Delta z_i$ are soil moisture (m$^3$ m$^{-3}$) and soil depth (cm) at $i$th layer, $N_{\mathrm{root}}$ and $z_{\mathrm{root}}$ are total
number of soil layers with roots and total depth (cm) of root zone, respectively.
Following the Ball-Berry stomatal resistance scheme, the $\theta$-controlled $\beta$ factor and other multiplicative factors
including temperature, foliage nitrogen simultaneously determine the maximum carboxylation rate $V_{\mathrm{max}}$ as follows:
$$V_{\mathrm{max}} = V_{\mathrm{max25}} \, \alpha_{\mathrm{vmax}}^{\frac{T_{\mathrm{v}}-25}{10}} \, f(N) f(T_{\mathrm{v}}) \, \beta \tag{2}$$

where $V_{\mathrm{max25}}$ is maximum carboxylation rate at 25 ℃ ($\mu$mol CO$_2$ m$^{-2}$ s$^{-1}$); $\alpha_{\mathrm{vmax}}$ is a parameter sensitive to vegetation
canopy surface temperature $T_{\mathrm{v}}$; $f(N)$ is a factor representing foliage nitrogen and $f(T_{\mathrm{v}})$ is a function that mimics thermal
breakdown of metabolic processes. Based on $V_{\mathrm{max}}$, carboxylase-limited (Rubisco limited) and export-limited (for C3
plants) photosynthesis rates per unit LAI ($A_C$ and $A_S$ respectively) are calculated, and the minimum of $A_C$, $A_S$ and light-
limited photosynthesis rates determine stomatal conductance $r_{\mathrm{s}}$, and, consequently, the ET over vegetated areas. For
the complete NOAHMP configuration, please see Table S1 in the supplementary material.





CLSM simulates the 0–2 and 0–100 cm soil moisture and evaporative stress as a function of simulated $\theta$ and
environmental variables. ET is then estimated based on the estimated evaporative stress and land-atmosphere humidity
gradients. Energy and water flux estimates are iterated with soil state estimates (e.g., $\theta$ and soil temperature) to ensure
closure of surface energy and water balances. For the detailed explanation of CLSM, please refer to Koster et al.

148 (2000).

GLEAM is a set of algorithms dedicated to the estimation of terrestrial ET and root-zone $\theta$ from satellite data. In this
study, the latest version of this model (v3.2a) is employed. In GLEAM, the configuration of soil layers varies as a
function of the land-cover type. Soil stratification is based on three soil layers for tall vegetation (0–10, 10–100, and
100–250 cm), two layers for low vegetation (0–10, 10–100 cm) and only one layer for bare soil (0–10 cm) (Martens
et al., 2017).
The cover-dependent PET (mm day$^{-1}$) of GLEAM is calculated using the Priestley and Taylor (1972) equation based
on observed air temperature and net radiation. Following this, estimates of PET were converted into actual
transpiration or bare soil evaporation (depending on the land-cover type, ET (mm day$^{-1}$)), using a cover-dependent,
multiplicative stress factor $S$ (–), which is calculated as a function of microwave vegetation optical depth (VOD) and
root-zone $\theta$ (Miralles et al., 2011). The related expressions are as follows:
$$\text{ET} = \text{PET} \times S + E_i \tag{3}$$

$$S = \sqrt{\frac{\text{VOD}}{\text{VOD}_{\text{max}}}\left(1 - \left(\frac{\theta_c - \theta\omega}{\theta_c - \theta_{\text{wilt}}}\right)^2\right)} \tag{4}$$

where $E_i$ is rainfall interception (mm); $S$ essentially represents the fPET (see Sect. 2.3) estimated by GLEAM; $\theta_c$ (m$^3$
m$^{-3}$) is the critical soil moisture and $\theta_\omega$ (m$^3$ m$^{-3}$) is the soil moisture content of the wettest layer, assuming that plants
withdraw water from the layer that is most accessible. Based on (4), GLEAM $S$ (or fPET) tend to become more
sensitive to $\theta$ in areas of low VOD seasonality (i.e., low differences between VOD and VOD$_{\text{max}}$). As for bare soil
conditions, $S$ is linearly related to surface soil moisture ($\theta_1$):
$$S = 1 - \frac{\theta_c - \theta_1}{\theta_c - \theta_{\text{wilt}}}. \tag{5}$$

To resolve variations in the vertical discretization of $\theta$ applied by each model, we linearly interpolated NOAHMP,
CLSM and GLEAM outputs into daily 0–10 and 0–40 cm soil moisture values using depth-weighted averaging.

**2.3 Variable indicating soil moisture and surface flux coupling**

Soil moisture – ET coupling can be diagnosed using a variety of different variables derived from ET, e.g. the fraction
of PET (fPET, the ratio of ET and PET) or the evaporative fraction (EF, the ratio of LE and the sum of LE and sensible
heat). Since ET is strongly tied to net radiation (Rn) (Koster et al., 2009), both fPET and EF are advantageous in that





they normalize ET and removing the impact of non-soil moisture influences on ET (e.g., net radiation, wind speed and
soil heat flux (G)). However, since sensible heat flux is not provided in the GLEAM dataset, we are restricted here to
using fPET.
It should be noted that the applied meteorological forcing data for NOAHMP and CLSM were somewhat different
from those used for GLEAM. Therefore, to minimize the impact of this difference, NOAHMP and CLSM fPET were
computed from North American Regional Reanalysis (NARR) using the modified Penman scheme of Mahrt and Ek
(1984) while GLEAM fPET was calculated using its own internal PET estimates. To examine the impact of PET
source, AmeriFlux fPET calculations were calculated using both GLEAM- and NARR-based PET values.
**2.4 Information measures**
Mutual information (MI) (Cover and Thomas, 1991) is a nonparametric measure of correlation between two random
variables. MI and the related Shannon-type entropy (Shannon, 1948) are calculated as follows. Entropy about a random
variable $\zeta$ is a measure of uncertainty according to its distribution $p_\zeta$ and is estimated as the expected amount of
information from $p_\zeta$ sample:
$$\mathrm{H}(p_\zeta) = \mathrm{E}_\zeta[-\ln(p_\zeta(\zeta))]. \tag{6}$$
Likewise, MI between $\zeta$ and another variable $\psi$ can be thought of as the expected amount of information about variable
$\zeta$ contained in a realization of $\psi$ and is measured by the expected Kullback-Leibler (KL) divergence (Kullback and
Leibler, 1951) between the conditional and marginal distributions over $\zeta$:
$$\mathrm{MI}(\zeta;\psi) = \mathrm{E}\psi[D(p_{\zeta\mid\psi} \| p_\zeta)]. \tag{7}$$
In this context, the generic random variables $\zeta$ and $\psi$ represent fPET and $\theta$ (soil moisture) respectively. The observation
space of the target random variable fPET was discretized using a fixed bin width. As bin width decreases, entropy
increases but mutual information asymptotes to a constant value. On the other hand, increased bin width requires more
sample size, which cannot always be satisfied. The trick is choosing a bin width where the NMI values stabilize with
sample size. After careful sensitivity analysis, we choose a fixed bin width of 0.25 [-] for fPET and make sure that
each AmeriFlux site have enough samples to accurately estimate the NMI, and change of this constant bin width from
0.1–0.5 [-] will not significantly alter our conclusions. Following Nearing et al. (2016), a bin width of 0.01 m$^3$ m$^{-3}$ (1%
volumetric water content) for $\theta$ was applied. Integrations required for MI calculation in Eq. (7) are then approximated
as summations over the empirical probability distribution function bins (Paninski, 2003).
By definition, the MI between two variables represents the amount of entropy (uncertainty) in either of the two
variables that can be reduced by knowing the other. Therefore, the MI normalized by the entropy of the AmeriFlux-
based fPET measurements represents the fraction of uncertainty in fPET that is resolvable given knowledge of the soil
moisture state (Nearing et al., 2013). Unlike Pearson's correlation coefficient, MI is insensitive to the impact of





nonlinear variable transformations. Therefore, it is well suited to describe the strength of the (potentially non-linear)
relationship between $\theta$ and fPET.
Here, we applied this approach to calculate the MI content between soil moisture representing different vertical depths
(as reflected by $\theta_S$ and $\theta_V$) and fPET at each AmeriFlux site. All estimated site-specific MI were normalized by the
entropy of the corresponding AmeriFlux-based fPET measurements to remove the effect of inter-site entropy
variations on the magnitude of NMI differences. The resulting normalized MI calculations between both $\theta_S$ and $\theta_V$
and fPET are denoted as NMI($\theta_S$, fPET) and NMI($\theta_V$, fPET) respectively.
The underestimation of observed $\theta$/ET coupling via the impact of mutually-independent $\theta$ and ET errors in AmeriFlux
observations (Crow et al. 2015) was minimized by focusing on the ratio between NMI($\theta_S$, fPET) and NMI ($\theta_V$, fPET).
To quantify the standard error of NMI differences between various soil moisture products, we applied a nonparametric,
500-member bootstrapping approach, and calculated pooled average of sampling errors across all sites assuming
spatially independent sampling error.
Finally, we also examined the impact of potential nonlinearity in the θ/ET relationship by comparing non-parametric
NMI results with comparable inferences based on a conventional Pearson's correlation calculation. The correlation-
based coupling strength between $\theta_S$ and fPET was denoted as $R(\theta_S$, fPET) and between $\theta_V$ and fPET as $R(\theta_V$, fPET).
**3 Results**
**3.1 Comparison of NMI($\theta_S$, fPET) and NMI($\theta_V$, fPET)**
Figure 1 contains boxplots of modelled and observed NMI($\theta_S$, fPET) and NMI($\theta_V$, fPET), i.e., the relative magnitude
of fPET information contained in surface soil moisture and vertically-integrated (0–40 cm) soil moisture, sampled
across all the AmeriFlux locations listed in Table 1. According to the AmeriFlux ground measurements, median values
of NMI($\theta_S$, fPET) and NMI($\theta_V$, fPET) (across all sites) are near 0.3 [-]. This suggests that approximately 30% of the
uncertainty (i.e., entropy at this particular bin width of 0.25 [-]) in fPET can be eliminated given knowledge of either
surface or vertically integrated soil moisture state. This is consistent with earlier results in Qiu et al., (2016) who used
similar variables to evaluate $\theta$/EF (evaporative fraction) coupling strength. The sampled medians of NMI($\theta_S$, fPET)
and NMI($\theta_V$, fPET) estimated by the NOAHMP and CLSM models are similar to these (observation-based) AmeriFlux
values. With the single exception that CLSM predicts much larger site-to-site variation in NMI($\theta_S$, fPET).
In contrast, NMI($\theta_S$, fPET) and NMI($\theta_V$, fPET) values sampled from GLEAM $\theta$ and fPET estimates are biased high
(with median NMI($\theta_S$, fPET) and NMI($\theta_V$, fPET) values of about 0.5 and 0.4 [-], respectively) with respect to all other
estimates.



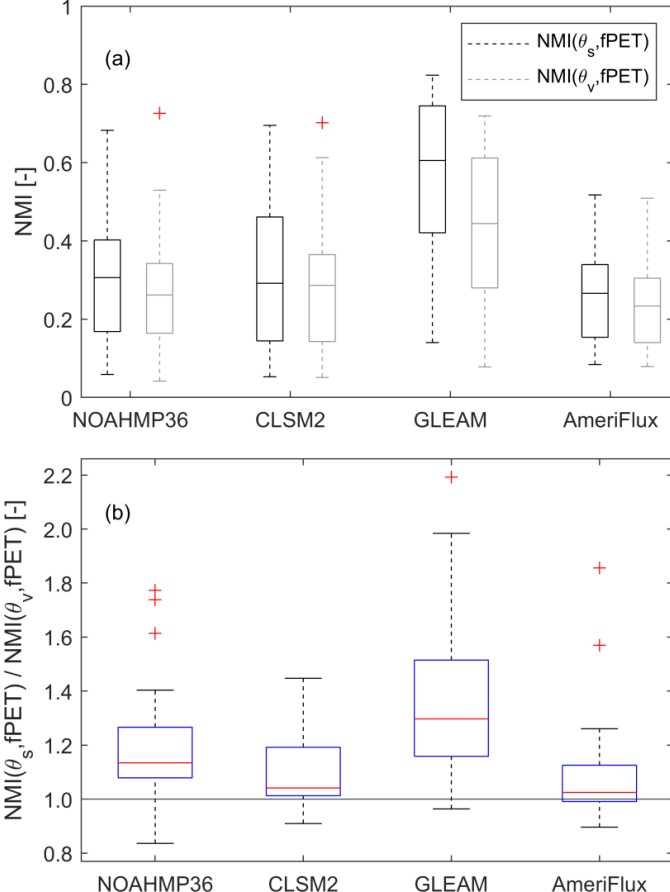

Fig.1 The $\theta$/ET coupling strengths for summertime anomaly time series acquired from various LSMs and AmeriFlux measurements:
(a) NMI($\theta_S$, fPET) and NMI($\theta_V$, fPET) individually and (b) NMI($\theta_S$, fPET) normalized by NMI($\theta_V$, fPET).
All three LSMs overall exhibit significantly (at $p = 0.05$ [-] confidence, using the 34 AmeriFlux site-collocated samples
pixels for pair $t$-test) higher NMI($\theta_S$, fPET) compared to NMI($\theta_V$, fPET) – implying the surface soil moisture
observation contain more fPET information than vertically-integrated soil moisture. However, the difference between
NMI($\theta_S$, fPET) and NMI($\theta_V$, fPET) is less discernible in AmeriFlux measurements (Fig. 1(a)).
Here, AmeriFlux observations are used as a baseline for LSM evaluation. However, it should be stressed that random
observation errors in $\theta$ and fPET will introduce a low bias into AmeriFlux-based estimates of both NMI($\theta_S$, fPET) and
NMI($\theta_V$, fPET) (Crow et al., 2015) and thus their difference as well. To address this concern, Fig. 1(b) plot the ratio
of NMI($\theta_S$, fPET) and NMI($\theta_V$, fPET), which normalizes and minimizes such observation error impacts. Ratio results
illustrate the general tendency for NMI($\theta_S$, fPET) > NMI($\theta_V$, fPET) discussed above. They also highlight the tendency
for GLEAM to overvalue $\theta_S$ (relative to $\theta_V$) when estimating fPET. A second approach for reducing the random error
of $\theta$ and fPET measurement errors is the Triple Collocation (TC)-based correction applied in Crow et al. (2015).


However, this approach is currently restricted to linear correlation and cannot be applied to NMI. Future work will
examine extending the information-based TC approach of Nearing et al. (2017), to the examination of NMI.

**3.2 Sensitivity of AmeriFlux-based NMI($\theta_S$, fPET)/NMI($\theta_V$, fPET)**

As mentioned in Sect. 2.1, an important concern is the impact of interpolation errors used to estimate 0–40 cm $\theta_V$ from
AmeriFlux $\theta_S$ observations acquired at non-uniform depths. To ensure that different methods for calculating
AmeriFlux $\theta_V$ values do not affect the main conclusion of this study, we configured three cases for $\theta_V$ calculation, and
compared their NMI($\theta_S$, fPET)/NMI($\theta_V$, fPET) results in Fig. 2. Case I reflects the baseline use of the exponential
filter described in Sect. 2.1. However, slight changes to AmeriFlux results are noted if alternative approaches are used.
Specifically, AmeriFlux-based NMI($\theta_V$, fPET) increases and closes the gap with NMI($\theta_S$, fPET) if the bottom-layer
soil moisture measurements are instead directly used as $\theta_V$ (Case II) or if 0–40 cm $\theta_V$ is based on the linear interpolation
of the two AmeriFlux $\theta$ observations (Case III), the impact of this modest sensitivity on key results is discussed below.

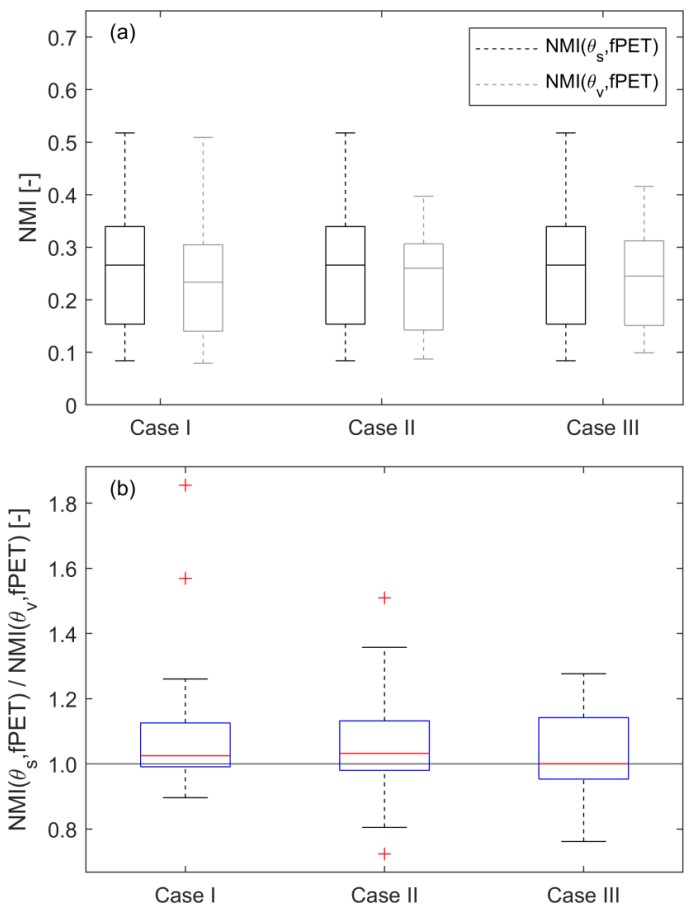



Fig.2 The $\theta$/ET coupling strengths for summertime anomaly time series from AmeriFlux measurements using three different $\theta_V$
calculation methods: (a) NMI($\theta_S$, fPET) and NMI($\theta_V$, fPET) individually and (b) NMI($\theta_S$, fPET) divided by NMI($\theta_V$, fPET) for
multiple $\theta_V$ cases. Case I is based on the application of an exponential filter to extrapolate 0–10 cm $\theta_S$ to a consistent 0–40 cm
bottom layer depth, while Cases II and III refer to the direct use of only the bottom layer measurement and a linear interpolation of
both the top and bottom layer, respectively, to calculate $\theta_V$ (see Sect. 2.1 for details on each case).
In addition, switching from GLEAM- to NARR-based PET when calculating fPET for AmeriFlux-based NMI($\theta_S$,
fPET) and NMI($\theta_V$, fPET) does not qualitatively change results and produces only a very slight (~6%) increase in the
median NMI($\theta_S$, fPET)/NMI($\theta_V$, fPET) ratio.

### 3.3 Spatial distribution of NMI($\theta_S$, fPET) and NMI($\theta_V$, fPET)

Figure 3 plots the spatial distribution of NMI($\theta_S$, fPET) and NMI($\theta_V$, fPET) results for each of the individual 34
AmeriFlux sites listed in Table 1. The climatic regime is represented by AI (aridity index) plotted as the background
color in Fig. 3. It can be seen in Fig. 3 that NMI($\theta_S$, fPET) estimates from LSMs are spatially related to hydro-climatic
conditions, as NOAHMP and CLSM predict that $\theta_S$ is moderately coupled with fPET (NMI($\theta_S$, fPET) of 0.3–0.5 [-])
in the arid Southwestern US (AI<0.2) and only loosely coupled with fPET in the relatively humid Eastern US. A
similar decreasing trend of NMI($\theta_S$, fPET) from the Southwestern to Eastern US is also captured by GLEAM.
However, as noted above, GLEAM generally overestimates NMI($\theta_S$, fPET) and NMI($\theta_V$, fPET) compared to
NOAHMP, CLSM and AmeriFlux. In contrast, a relatively weaker spatial pattern emerges in AmeriFlux-based
NMI($\theta_S$, fPET) results. In addition, spatial patterns for NMI($\theta_S$, fPET) are less defined than for NMI($\theta_V$, fPET) in all
four datasets.
Scatterplots in Fig. 4 summarize the spatial relationship between LSM-based NMI($\theta_S$, fPET) and NMI($\theta_V$, fPET)
results versus AmeriFlux observations. While observed levels of correlation in Fig. 4 are relatively modest, there
appears to be a significant level of spatial correspondence between modelled and observed NMI results – motivating
the need to better understand processes responsible for spatial variations in NMI results.





Fig. 3 NMI($\theta_S$, fPET) (left column) and NMI($\theta_V$, fPET) (right column) estimates at AmeriFlux sites for: (a) NOAHMP, (b) CLSM, (c) GLEAM and (d) AmeriFlux. Marker color reflects NMI magnitudes and symbol type reflects local land cover type at each site. Background color shading reflects aridity index (AI) values.



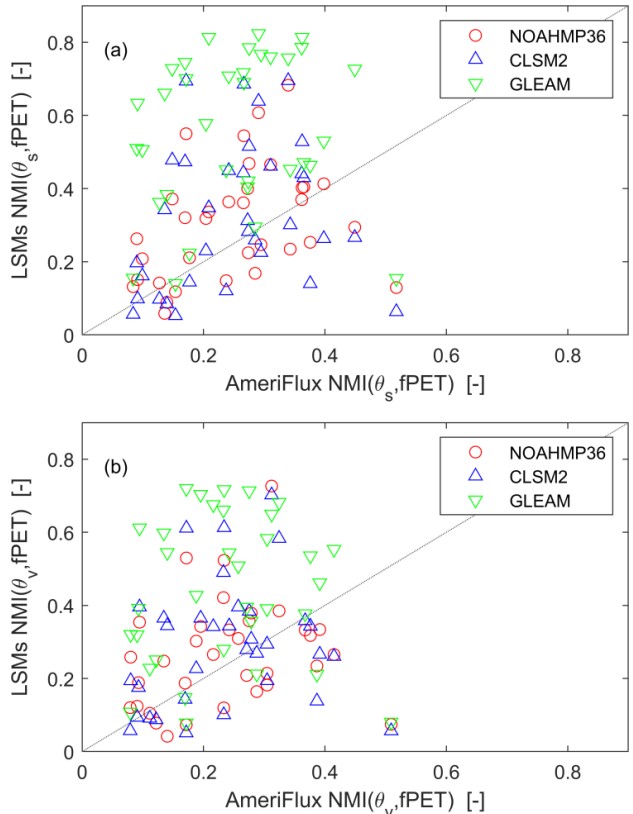


Fig. 4 Scatterplot of LSM-based (a) NMI($\theta_S$, fPET) and (b) NMI($\theta_V$, fPET) results versus AmeriFlux observations.
**3.4 Sensitivity of NMI($\theta_S$, fPET)/NMI($\theta_V$, fPET) ratio to climatic conditions**
Figure 5 further summarizes the ratio of NMI($\theta_S$, fPET) and NMI($\theta_V$, fPET) as a function of AI for all four products
(NOAHMP, CLSM, GLEAM and AmeriFlux). Error bars represent the standard deviation of sampling errors
calculated from a 500-member bootstrapping analysis. With increasing AI, there is a decreasing trend in surface and
vertically integrated $\theta$/ET coupling within both NOAHMP and CLSM. This decreasing trend is particularly clear when
AI is below 1.0 [-]. NOAHMP, CLSM and GLEAM estimates of NMI($\theta_S$, fPET) are generally higher than NMI($\theta_V$,
fPET) in all climatic conditions. There is relatively lower sensitivity to aridity captured in the AmeriFlux
measurements, as the NMI($\theta_S$, fPET)/NMI($\theta_V$, fPET) ratio still approximates one under semiarid conditions (i.e., AI
< 0.5 [-]).
Connecting these findings to spatial distribution of NMI($\theta_S$, fPET) and NMI($\theta_V$, fPET) (Fig. 3), it is confirmed that
the relative magnitudes of NMI($\theta_S$, fPET) and NMI($\theta_V$, fPET) for all three LSMs are spatially related to hydro-climatic
regimes (although in fundamentally different ways). In contrast, this link is weaker in the AmeriFlux measurements
which, except for a small fraction of very low AI sites, do not appear to vary as a function of AI. These conclusions





are not qualitatively impacted by looking at NMI($\theta_S$, fPET) and NMI($\theta_V$, fPET) differences, as opposed to their ratio
as in Fig. 5, or by looking at $R(\theta_S$, fPET) and $R(\theta_V$, fPET) instead of NMI.

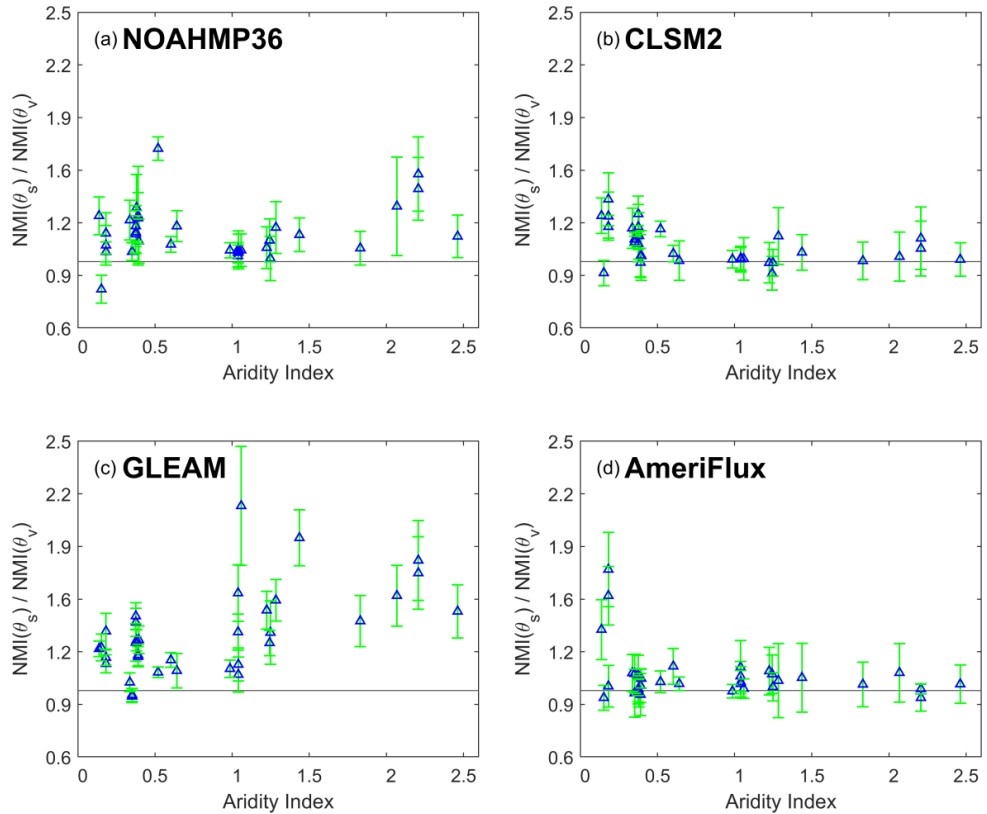


Fig. 5 For a) NOAHMP, (b) CLSM, (c) GLEAM and (d) AmeriFlux estimates, the ratio of NMI($\theta_S$, fPET) and NMI($\theta_V$, fPET) as
a function of AI across all AmeriFlux sites.

**4 Discussion and conclusion**

Since transpiration dominates the global ET (Jasechko et al., 2013), deep-layer soil moisture ($\theta_V$) is generally
considered to contain more ET information than that of surface soil moisture ($\theta_S$) – given plant transpiration is
balanced by root water uptake from deeper soils (Seneviratne et al., 2010). However, this assumption is rarely tested
using models and/or observations. Here, we apply normalized mutual information (NMI) to examine how the vertical
support of a soil moisture product impacts its relationship with concurrent surface ET.
Specifically, using AmeriFlux ground observations, we examine whether (NMI-based) estimates of LSM $\theta_S$ versus
ET and $\theta_V$ versus ET coupling strength accurately reflect observations acquired at a range of AmeriFlux sites. In
general, compared to the baseline case of exponential filter extrapolated 40-cm bottom layer $\theta_V$, LSMs agree with
AmeriFlux observations in that the overall fPET information contained in $\theta_S$ is slightly higher than that of $\theta_V$ (Fig. 1).


However, the sensitivity analysis showed this difference between NMI($\theta_S$, fPET) and NMI($\theta_V$, fPET) diminishes when
using different methods for calculating $\theta_V$ using AmeriFlux observations (Fig. 2). As a result, this result should be
viewed with caution.
While NOAHMP and CLSM derived NMI($\theta_S$, fPET) and NMI($\theta_V$, fPET) results are generally consistent with the
AmeriFlux observations, GLEAM overestimates NMI($\theta_S$, fPET), NMI($\theta_V$, fPET), and the ratio NMI($\theta_S$,
fPET)/NMI($\theta_V$, fPET) relative to observations. Although both LSMs and GLEAM are based on the same classical
two-section (soil moisture-limited and energy-limited) ET regimes framework (Sect. 2.2), they differ in two
fundamental aspects. First, the evaporative stress is represented as a more direct and strong function of soil moisture
in GLEAM - see Eqs. (4) and (5) - which leads to the overestimation of $\theta$/ET coupling strength. This is consistent
with our results that GLEAM generally overestimates NMI($\theta_S$, fPET) and NMI($\theta_V$, fPET) consistently across all land
covers, compared to AmeriFlux-based estimates.
On the other hand, NOAHMP and CLSM approximate ET in the manner of biophysical models, and expresses
biophysical control on ET through the stomatal resistance $r_s$, which is a function of multiple limiting factors including
$\theta$. Therefore, the more complex ET scheme employed by NOAHMP and CLSM would seem to mitigate the
overestimation of NMI($\theta_S$, fPET) and NMI($\theta_V$, fPET), as other relevant factors besides $\theta$ (such as temperature, foliage
nitrogen) are also considered in determining maximum carboxylation rate $V_{max}$ and stomatal resistance $r_s$ - and
consequently more realistic actual ET. Secondly, the stress factor $\beta$ in both LSMs considers the cumulative effects of
$\theta$ conditions along different layers (Eq. (1)), while the corresponding $S$ factor in GLEAM only uses the wettest soil
layer condition, which is top layer at most sites. This likely explains the overestimation of the NMI($\theta_S$, fPET)/NMI($\theta_V$,
fPET) ratio by GLEAM.
Although the median values of NMI($\theta_S$, fPET) and NMI($\theta_V$, fPET) predicted by NOAHMP and CLSM are general in
line with AmeriFlux observations, they are more spatially related to hydro-climatic conditions (as summarized by AI)
than their counter parts acquired from AmeriFlux measurements. Seen from the plot of NMI($\theta_S$, fPET)/NMI($\theta_V$, fPET)
ratio as a function of AI (Fig. 5), the modelled and observed NMI($\theta_S$, fPET)/NMI($\theta_V$, fPET) ratio median decreases
with increasing AI, and the decreasing trend is particularly clear when AI is lower than 1.0 [-]. In contrast, there is
relatively lower sensitivity to aridity exhibited in the AmeriFlux measurements.
These results provide several key insights into future land-atmosphere coupling analysis and LSM development. First,
all the datasets – both model-based and ground-observed – indicates that $\theta_S$ contain at least as much ET information
as $\theta_v$. Hence, remote-sensing land surface soil moisture datasets are suitable, and should be considered, for analyzing
the general interaction between land and atmosphere, e.g., soil moisture – air temperature coupling (Dong and Crow,
2019) and the interplay of soil moisture and precipitation (Yin et al., 2014). Additionally, future generations of
GLEAM may consider more sophisticated evaporation stress functions, which may improve its accuracy in
representing soil's control on local ET. This may, in turn, improve the accuracy of GLEAM ET product. Finally, our
results demonstrate that modeled $\theta$/ET is highly sensitive to hydro-climates, compared to observed relationships.





Modifying the model structures to reduce such sensitivity might be necessary for accurately representing the
interaction of land surface and atmosphere across different climate zone. This may lead to more realistic projections
of future drought-induced heatwaves, when coupled with general circulation models.
**Data availability**
Ground-based soil moisture and surface flux data are available from http://ameriflux.ornl.gov/. GLEAM dataset is
available from https://www.gleam.eu/. LSMs simulations of NOAHMP and CLSM used in this study are available by
contacting the authors.
**Author contributions**
Jianxiu Qiu and Wade T. Crow conceptualized the study. Jianzhi Dong helped preparing the LSMs simulation. Grey S.
Nearing assisted in the mutual information analysis. Jianxiu Qiu carried out the analysis and wrote the first draft
manuscript, and Wade T. Crow refined the work. All authors contributed to the analysis, interpretation and writing.
**Competing interests**
The authors declare that they have no conflict of interest.
**Acknowledgments**
This work was supported by National Natural Science Foundation of China (Grant Nos. 41501450, 51779278) and
Natural Science Foundation of Guangdong Province, China (Grant No. 2016A030310154).

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
