# Peer review of "Model Representation of the Coupling between"

_Hydrology and Earth System Sciences, 2019_

## Short Comment (SC1) · 18 Jul 2019

Dear authors,

First, I would like to thank you for your advice to us. I appreciate it.

Indeed, simpler models are expected to condense information within fewer inter-variable relationships. By doing so, the outputs tend to become more sensitive to the inputs that remain considered; in other words, mutual information between the remaining variables increases. Taking GLEAM as an example — which I would recommend

the authors not to call a 'land surface model' but a 'remote sensing evaporation model', or similar (see literature on this) — the model follows the assumption that the main constraints on potential evaporation can be captured by the soil moisture signal. This implies that the relation between soil moisture and evaporation in GLEAM will be tighter than in nature; in reality, multiple other variables constrain stomatal conductance that are either neglected or implicitly assumed to co-vary with soil moisture. As CLSM and Noah use a more inclusive and comprehensive Ball–Berry model, their sensitivity to soil moisture is lower and likely closer to nature.

In my view, a more relevant question — especially for an evaporation retrieval model — is whether this simplification is valid in terms of the accuracy of the final evaporation output.

My first advisor, once told me that simpler models should be prioritized if their output is equally accurate. Getting the right results for the right reasons is in fact crucial for land surface models when they are designed to make future projections in a non-stationary environment, but not for retrieval methods like GLEAM that simply aim to produce an accurate historical data record. For the latter, what arguably matters the most is whether the selected input variables are observable. There again, the importance of making a distinction regarding the type of model and their purpose in the paper.

If the authors had chosen to look at the sensitivity of evaporation to radiation in-stead, they would have encountered very similar findings: because GLEAM uses a Priestley and Taylor simplification, the output will be more sensitive to radiation than in land surface models. This is again a given. The real question is whether the accuracy of the output is significantly affected by the simplification being taken. Like for the soil moisture stress function, this simplification also has some reasoning behind: other inputs required by Monteith's formulation (e.g., wind, vapour pressure deficit) are not

easily available from remote sensing. This for an evaporation retrieval model like GLEAM is crucial, but unimportant for land surface models like Noah or CLSM which are designed to run on climate model input or reanalysis.

Consequently, I have the feeling that the story would be more compelling if it had embraced a discussion on the degree these models can be simplified and still be comparable in terms of outputs. In that sense, for instance, I would advise to take a step back and show the in situ versus model validation results of ET and soil moisture. The authors might perhaps find that GLEAM performs worse than the land surface models. However, if they find the opposite, I would rather call this a success story on how simpler formulations can outperform more complex models, rather than merely highlighting that the physics in simpler models are more rudimentary, which is something we can all agree with.

Good luck with the review.

Best regards,
Diego

---

## Referee Comment (RC1) · Anonymous Referee #1 · 29 Aug 2019

Evapotranspiration is one of the most important land flux extracting water from the soil. Its estimation is particularly challenging and many different type of techniques have been used which utilize either energy balance or coupled energy and water balance also including elements of the carbon cycle. Several ET estimation techniques have already been developed which differ in their data requirements, the approaches used to derive them and their estimates (simplified models, remote sensing and so on), yet it is not clear which provides the most reliable estimates. For models the main issues concern the different assumptions and simplifications made which can significantly alter the final ET estimates. For these reasons the present study is important and needs to be considered for potential publication on HESS.

The manuscript is well written and well structured. It does not contain significant issues. I only have one comment for one issue which I think the authors should consider. In the study I did not find any particular discussion related to the type of vegetation characterizing the Ameriflux sites and its effect on the result. I think that vegetation type can be relatively important as for example grass roots are shallower with respect to tree and shrub roots and thus can exert potential different effects both on the coupling strength between the soil moisture profile (surface vs root zone) and on the transpiration flux itself also considering that transpiration is the dominant pathway for the total Evapotranspiration and is estimated to account for two-thirds of global land ET based on flux tower measurements (Schlesinger and Jasechko,2014). Based on that the authors should provide at least a discussion on the potential effects of the vegetation type on the presented results.

I have included additional comments on the annoted pdf.

Schlesinger, William H., and Scott Jasechko. "Transpiration in the global water cycle." Agricultural and Forest Meteorology 189 (2014): 115-117.

Please also note the supplement to this comment:
https://www.hydrol-earth-syst-sci-discuss.net/hess-2019-282/hess-2019-282-RC1-supplement.pdf

**Supplement:**

[revised manuscript text omitted]

---

## Referee Comment (RC2) · Anonymous Referee #2 · 13 Oct 2019

Manuscript Number: HESS-2019-282 (Jianxiu et al)

I have reviewed this manuscript and I have the following comments.

EVALUATION

This is a well written paper with a clear contribution to ecohydrological modeling and I have very few comments. The first relates to the jargon in the title. Please try to simplify the title for the paper to be appealing to a wider audience. Secondly, the aims and objectives of the paper must be clearly formulated and also indicate what is new or novel about this study and who benefits from it? Lastly, what is the take-home message from this study given that no conclusions are given?

SPECIFIC COMMENTS

- Keywords: - "surface evapotranspiration" is listed as a keyword/phrase. Delete the word "surface" - Line 27 – indicate that some if the incoming energy is absorbed by the surface... given that you are mentioning biochemical cycles in line 30 - There are inconsistencies throughout the paper regarding the evaporation terms. A typical example is in lines 11 to 12 in the abstract where the authors refer to the sensible heat flux and evapotranspiration (ET) in the same sentence. Rather also use the energy equivalent of ET (i.e. the latent heat flux) and be consistent throughout the paper. - Line 59: What is meant by ET entropy? This is not a standard micrometeorology or ecohydrological phrase. Please define such terms. - Throughout the paper rather use the phrase "soil water content" which is more specific than "soil moisture" - Lines 63-64 not necessary - Line 75 sounds rather cyclic, rephrase! - How did you account for the accuracy of the different types of soil water content sensors or their depth of installation across the Ameriflux sites? How does this affect your results? - The vegetation acts as the link between the atmosphere and soil water content in deep soil profiles. Please give more details on how the vegetation types affected your analysis/results. - Line 188: What is the bottom layer soil moisture measurement? Define this, else rephrase. - 2) options for $\theta$ factor for stomatal resistance (the $\beta$ factor). Not clear what this represents. What is a theta factor? What does it do? - and reference soil moisture (m3 m$-$3), How is this defined? - and reference soil moisture (m3 m$-$3), Confusion over symbols. Sometimes you mention stomatal resistance, and at other times stomatal conductance; line 142. Choose one and stick to it otherwise this easily gets very confusing. - line 142 – stomatal conductance is not the sole driver of ET. Its more complex than that. Please elaborate - Eqn 6: what does the symbol H mean here? Thought you said H was the sensible heat flux earlier? - Fig 4 these are poor model performances.

Comments end.

---

## Author Comment (AC1) · 15 Oct 2019

We would like to thank Dr. Diego G. Miralles for his constructive comments. We are generally in agreement with his sentiments. Most importantly, we agree that our original manuscript was overly aggressive in lumping various evapotranspiration (ET) estimation approaches into a single conceptual category. As Diego points out, there are important differences between these approaches that are relevant for the stated purposes of our paper.

Nevertheless, we would like to stress that all approaches considered in our paper contain (at their core) a parameterized relationship between soil moisture (SM) and evapotranspiration (ET). While the implications of mis-parameterization this relationship are arguably more severe for a land surface model, we'd argue that the issue remain relevant for any approach (such as GLEAM) that utilizes a water balance (and/or data assimilation system) approach to estimate SM and, in turn, uses SM to constrain ET. Regardless of the complexity that a given approaches employs, failing to accurately describe the relationship between ET and (large number of potential) environmental constraints should eventually degrade the robustness of the model. We'd argue that this is true regardless of whether a model is employed as a retrospective, diagnostic or predictive manner. Our paper is an attempt to "open the lid" on these models to measure internal SM/ET coupling and explore the impact of potential mis-coupling on ET estimation.

Given this emphasis, Diego's suggestion to expand our analysis to include direct flux validation is an excellent one. Indeed, preliminary results suggest that, despite its simplicity, GLEAM does not underperform more complex land surface models with respect to daily ET predictions. Therefore, as Diego points out, any criticism of GLEAM must be tempered by this bottom-line result.

Therefore, if given the opportunity, we'd make the following changes to our current manuscript:

1. Change the characterization of GLEAM from a "land surface model" to "retrieval algorithm" in the revised manuscript and rather a more complete discussion of differences in complexity and envisioned application for various modelling approaches. However, regardless of how we characterize GLEAM, stress that is remains valuable to understand if its ET predictions respond to environmental factors (like soil moisture) accurately.

2. Look directly at the ET accuracy issue and better describe the connection between accurate coupling and the absolute accuracy of GLEAM ET predictions.

282, 2019.

---

## Author Comment (AC2) · 22 Oct 2019

The comment was uploaded in the form of a supplement:
https://www.hydrol-earth-syst-sci-discuss.net/hess-2019-282/hess-2019-282-AC2-supplement.zip

---

## Author Response (AR1)

**Reply to Dr. Diego G. Miralles interactive comment**

We would like to thank Dr. Diego G. Miralles for his constructive comments. We are generally in agreement with his sentiments. Most importantly, we agree that our original manuscript was overly aggressive in lumping various evapotranspiration (ET) estimation approaches into a single conceptual category. As Dr. Diego points out, there are important differences between these approaches that are relevant for the stated purposes of our paper.

Nevertheless, we would like to stress that all approaches considered in our paper contain (at their core) a parameterized relationship between soil water content ($\theta$) and ET. While the implications of mis-parameterization this relationship are arguably more severe for a land surface model, we believe that this issue remains relevant for any approach (such as GLEAM) that utilizes a water balance (and/or data assimilation system) approach to estimate $\theta$ and, in turn, uses $\theta$ to constrain ET. Regardless of the complexity that a given approach employs, failing to accurately describe the relationship between ET and (large number of potential) environmental constraints should eventually degrade the robustness of the model. We believe that this is true regardless of whether a model is employed as a retrospective, diagnostic or predictive manner. Our paper is an attempt to "open the lid" on these models to measure internal $\theta$/ET coupling and explore the impact of potential mis-coupling on ET estimation.

Given this emphasis, Diego's suggestion to expand our analysis to include direct flux validation is an excellent one. Indeed, preliminary results suggest that, despite its simplicity, GLEAM does not underperform more complex land surface models with respect to daily ET predictions. Therefore, as Diego points out, any criticism of GLEAM must be tempered by this bottom-line result.

Therefore, we've made the following changes to our current manuscript:

1. Change the characterization of GLEAM from a "land surface model" to "retrieval algorithm" throughout the revised manuscript and add a more complete discussion of differences in complexity and envisioned application for various modelling approaches.

2. Directly evaluate the GLEAM ET accuracy and better describe the connection between accurate $\theta$/ET coupling and the absolute accuracy of GLEAM ET predictions. Pertinent revisions are presented in new Fig. 6 and the related discussions.

**Reply to Referee #1 interactive comment**

We would like to thank Referee #1 for the constructive comments.

I only have one comment for one issue which I think the authors should consider. In the study I did not find any particular discussion related to the type of vegetation characterizing the AmeriFlux sites and its effect on the result. I think that vegetation type can be relatively important as for example grass roots are shallower with respect to tree and shrub roots and thus can exert potential different effects both on the coupling strength between the soil moisture profile (surface vs. root zone) and on the transpiration flux itself also considering that transpiration is the dominant pathway for the total evapotranspiration and is estimated to account for two-thirds of global land ET based on flux tower measurements (Schlesinger and Jasechko,2014). Based on that the authors should provide at least a discussion on the potential effects of the vegetation type on the presented results.

Thanks for the comments. In order to minimize the effect of different root depths from different vegetation types on NMI($\theta_S$, fPET) and NMI($\theta_V$, fPET), we used exponential filter to extrapolate $\theta$ to a unified 40 cm bottom layer depth and find that the overall fPET information contained in $\theta_S$ is slightly higher than that of $\theta_V$. However, the difference between NMI($\theta_S$, fPET) and NMI($\theta_V$, fPET) diminishes when using different methods for calculating $\theta_V$ using AmeriFlux observations.

We've added more extensive discussion regarding the role of vegetation on key results in the revised manuscript. In particular, Fig. 4 has been newly expanded to better isolate the impact of vegetation type and the role of vegetation types is now directly addressed via new text appearing in Section 3.3 of the revised manuscript.

Furthermore, we showed the result of NMI($\theta_S$, fPET)/NMI($\theta_V$, fPET) ratio as a function of vegetation type in Fig. A1. The conclusion that the overall fPET information contained in $\theta_S$ is slightly higher than that of $\theta_V$ does not vary with vegetation types, although NMI($\theta_S$, fPET) is much higher than NMI($\theta_V$, fPET) in open shrubland and woody savannas.

For the rest comments annotated in the manuscript:
1. P6 Line 141. *Ac* and *As* not defined
We've made the following revision in Section 2.2 to clearly defined *Ac* and *As*:
*"Based on V$_{max}$, photosynthesis rates per unit LAI including carboxylase-limited (Rubisco limited, denoted by A$_C$) type and export-limited (for C3 plants, denoted by A$_S$) type are calculated respectively."*

2. P9 Line 211-215. Maybe a statement to point to section 3.1 is necessary here.

As suggested, we've added a statement to directly point to results starting from Section 3.1:

*"Therefore, relative comparisons between NMI($\theta_S$, fPET) and NMI($\theta_V$, fPET) are based on examining the size of their mutual ratio NMI($\theta_S$, fPET)/NMI ($\theta_V$, fPET)."*

3. P9 Line 222. Is it for Case I?

Yes, the "vertically-integrated (0–40 cm) soil moisture" is estimated from Case I. We've also clarified this in Section 3.1:

*"…i.e., the relative magnitude of fPET information contained in surface soil water content and vertically-integrated (0–40 cm) soil water content estimated from Case I…"*

4. P14 Line 287. Even though the sample size is small it would be nice to have also similar plots and the plots above for different vegetation type.

As suggested, we've revised Fig. 4 so that samples are plotted separately according to their vegetation types. With varying magnitudes, the overall overestimation of GLEAM is observed across different vegetation types.

5. P14 Line 291-294. This trend is not really evident. I see an evident increasing ratio only when AI approaches to zero. Maybe a statistical significance of this trend should analyzed.

As suggested, we've added analysis of statistical significance of this trend. Indeed, the increasing trend of NMI($\theta_S$, fPET)/NMI($\theta_V$, fPET) ratio is more evident for CLSM, with a moderate goodness-of-fit (0.28). We've also clarified this in Section 3.4:

*"With increasing AI, there is a significant decreasing trend in both NMI($\theta_S$, fPET) and NMI($\theta_V$, fPET) for all three simulations, with a goodness-of-fit above 0.5 (figure not shown). For all cases, the NMI($\theta_S$, fPET)/NMI($\theta_V$, fPET) ratios are consistently greater than unity under all climatic conditions. However, the estimated NMI($\theta_S$, fPET)/NMI($\theta_V$, fPET) ratios from all three simulations (NOAHMP, CLSM and GLEAM) exhibit quite different trends with respect to AI. The NMI($\theta_S$, fPET)/NMI($\theta_V$, fPET) ratio for CLSM decreases with increasing AI, with a moderate goodness-of-fit value of 0.28,…"*

6. P15 Line 315. This can also depend upon the vegetation type as grass and trees are characterized by different root depths. They can exert a different effects on the coupling between soil moisture and evapotranspiration.

Thanks for the comments. This concern of different root depths impact is addressed by applying different methods to retrieve vertically integrated $\theta$ as we stated in Section 2.1. The entire analysis is based on the default case I that exponentially filter $\theta$ to a unified 40 cm bottom layer depth, and it is found that the overall fPET information contained in $\theta_S$ is slightly higher than that of $\theta_V$. However, the difference between NMI($\theta_S$, fPET) and NMI($\theta_V$, fPET) is less obvious when using different methods for calculating $\theta_V$ using AmeriFlux observations.

In addition, we've showed the result of NMI($\theta_S$, fPET)/NMI($\theta_V$, fPET) ratio as a function of vegetation type in Fig. A1. The conclusion that the overall fPET information contained in $\theta_S$ is slightly higher than that of $\theta_V$ does not vary with vegetation types, although NMI($\theta_S$, fPET) is obviously higher than NMI($\theta_V$, fPET) in open shrubland and woody savannas.

**Reply to Referee #2 interactive comment**

We would like to thank Referee #2 for the constructive comments.

This is a well written paper with a clear contribution to ecohydrological modeling and I have very few comments. The first relates to the jargon in the title. Please try to simplify the title for the paper to be appealing to a wider audience. Secondly, the aims and objectives of the paper must be clearly formulated and also indicate what is new or novel about this study and who benefits from it? Lastly, what is the take-home message from this study given that no conclusions are given?

Thanks for the comments. We agree that our original title could be improved. Accordingly, the title of revised manuscript has been changed to "Model Representation of the Coupling between Evapotranspiration and Soil Water Content at Different Depths." We feel that this is more accessible to a broader audience.

In addition, we've revised the abstract and introduction to better emphasize the aim and objectives of the paper and provide a concise summary of major conclusion and the target readers with most potential interest are also highlighted in the abstract.

SPECIFIC COMMENTS
- Keywords: - "surface evapotranspiration" is listed as a keyword/phrase. Delete the word "surface"

Thank you for these comments. The keyword of "surface evapotranspiration" has been revised as suggested.

- Line 27 – indicate that some if the incoming energy is absorbed by the surface… given that you are mentioning biochemical cycles in line 30

To avoid this issue, we've removed all mentions of biochemical cycles in the manuscript.

- There are inconsistencies throughout the paper regarding the evaporation terms. A typical example is in lines 11 to 12 in the abstract where the authors refer to the sensible heat flux and evapotranspiration (ET) in the same sentence. Rather also use the energy equivalent of ET (i.e. the latent heat flux) and be consistent throughout the paper.

Thank you for this comment – we agree this was an issue in the original manuscript. In the revised version, the energy equivalent of ET (i.e., the latent heat flux) has been used consistently when also referencing sensible heat flux.

- Line 59: What is meant by ET entropy? This is not a standard micrometeorology or ecohydrological phrase. Please define such terms.

Thank you for the comments. The original expression of "corresponding ET entropy" refers to the entropy of a corresponding ET time series. This is clarified in the revised manuscript.

- Throughout the paper rather use the phrase "soil water content" which is more specific than "soil moisture"
We've replaced the expressions of "soil moisture" with "soil water content" throughout the manuscript.

- Lines 63-64 not necessary
These two unnecessary sentences have been removed as suggested.

- Line 75 sounds rather cyclic, rephrase!
The sentence has been rephrased to "*As described above, θ/ET coupling assessments made using AmeriFlux observations are compared with those using state-of-the-art LSMs including…*"

- How did you account for the accuracy of the different types of soil water content sensors or their depth of installation across the AmeriFlux sites? How does this affect your results?
As the most of the AmeriFlux sites involved in the analysis are using frequency domain reflectometer probe for soil water content measurements, the impact of different sensors on our conclusion is limited.

Secondly, to minimize the effect of different measurement depths on our analysis, we designed three different cases to estimate vertically integrated soil water content ($\theta_V$). Case I was based on the application of an exponential filter (Wagner et al., 1999; Albergel et al., 2008) to extrapolate $\theta_S$ to a consistent 40 cm bottom layer depth. Therefore, only $\theta_S$ was used to derive $\theta_V$ and the bottom-layer (or second layer) AmeriFlux $\theta$ measurement was neglected in this case. Nevertheless, since the quality of $\theta_V$ estimates is important in our analysis, we also calculated two additional cases where 0–40 cm $\theta_V$ was estimated using: 1) the bottom-layer soil water content measurement acquired at each AmeriFlux site (hereinafter, Case II) and 2) linear interpolation of $\theta_S$ and the bottom-layer AmeriFlux soil water content measurement (hereinafter, Case III).

The sensitivity of key results show that compared to the baseline Case I of exponential filter extrapolated 40-cm bottom layer $\theta_V$, LSMs and GLEAM agree with AmeriFlux observations in that the overall fPET information contained in $\theta_S$ is slightly higher than that of $\theta_V$. However, the sensitivity analysis showed this difference between NMI($\theta_S$, fPET) and NMI($\theta_V$, fPET) diminishes when using different methods for calculating $\theta_V$ using AmeriFlux observations. These experimental designs and the corresponding findings are clearly stated in the revised manuscript.

- The vegetation acts as the link between the atmosphere and soil water content in deep soil profiles. Please give more details on how the vegetation types affected your analysis/results.

Thanks for the comments. As mentioned in the response to the previous comment, in order to minimize the effect of different root depths from different vegetation types on NMI($\theta_S$, fPET) and NMI($\theta_V$, fPET), we used an exponential filter to extrapolate $\theta$ to a unified 40 cm bottom layer depth and find that the overall fPET information contained in $\theta_S$ is slightly higher than that of $\theta_V$. However, the difference between NMI($\theta_S$, fPET) and NMI($\theta_V$, fPET) diminishes when using different methods for calculating $\theta_V$ using AmeriFlux observations.

We've added more extensive discussion regarding the role of vegetation on key results in the revised manuscript. In particular, Fig. 4 has been newly expanded to better isolate the impact of vegetation type and the role of vegetation types is now directly addressed via new text appearing in Section 3.3 of the revised manuscript.

Furthermore, we showed the result of NMI($\theta_S$, fPET)/NMI($\theta_V$, fPET) ratio as a function of vegetation type in Fig. A1. The conclusion that the overall fPET information contained in $\theta_S$ is slightly higher than that of $\theta_V$ does not vary with vegetation types, although NMI($\theta_S$, fPET) is much higher than NMI($\theta_V$, fPET) in open shrubland and woody savannas.

- Line 107: What is the bottom layer soil moisture measurement? Define this, else rephrase.

As soil water content measurements are generally available at two discrete depths at the AmeriFlux sites, the bottom layer measurements refer to the measurements at the deeper depth or the second observation layer from surface. This has been clarified in the revised manuscript.

- 2) options for $\theta$ factor for stomatal resistance (the $\beta$ factor). Not clear what this represents. What is a theta factor? What does it do? - and reference soil moisture (m3 m-3), How is this defined? Confusion over symbols.

The $\theta$ factor stands for soil water content, and different expressions of $\theta$ lead to different representations of relationship between $\theta$ and stress factor $\beta$. We've revised the original expression to "*…and schemes controlling the effect of $\theta$ on the vegetation stress factor $\beta$*". As clarified in the revised manuscript, reference soil moisture is set as field capacity in the NOAH users' guide for parameterization.

- Sometimes you mention stomatal resistance, and at other times stomatal conductance; line 142. Choose one and stick to it otherwise this easily gets very confusing.

As suggested, we've revised the only occurrences of the term "stomatal conductance" in Section 2.2 into "stomatal resistance" to avoid any confusion.

- line142 – stomatal conductance is not the sole driver of ET. It's more complex than that.

To avoid such confusion, we've revised expression as "*The minimum of $A_C$, $A_S$ and*

*light-limited photosynthesis rate determines stomatal resistance $r_s$, and consequently affects ET over vegetated areas*".

- Please elaborate - Eqn 6: what does the symbol H mean here? Thought you said H was the sensible heat flux earlier?

In the original Eq. 6, H represents Shannon-type entropy of the variable $\zeta$. Indeed, it could be easily confused with sensible heat flux symbol mentioned in Section 1. Therefore, we've replaced the symbol H in Eq. 6 with SE.

- Fig 4 these are poor model performances.

Indeed, the consistency of NMI($\theta$, fPET) between models and observations varies across different vegetation types, and varies across different models. However, it should be noted that the absolute value of NMI($\theta$, fPET) is not a direct index to measure model performance. Furthermore, our analysis conclusion will not be affected as we are using the relative ratio 
[revised manuscript text omitted]